# A Systems Analysis of Phenotype Heterogeneity in APOE*3Leiden.CETP Mice Induced by Long-Term High-Fat High-Cholesterol Diet Feeding

**DOI:** 10.3390/nu14224936

**Published:** 2022-11-21

**Authors:** Yared Paalvast, Enchen Zhou, Yvonne J. W. Rozendaal, Yanan Wang, Albert Gerding, Theo H. van Dijk, Jan Freark de Boer, Patrick C. N. Rensen, Ko Willems van Dijk, Jan A. Kuivenhoven, Barbara M. Bakker, Natal A. W. van Riel, Albert K. Groen

**Affiliations:** 1Department of Pediatrics, University of Groningen, University Medical Center Groningen, 9713 AV Groningen, The Netherlands; 2Department of Medicine, Division of Endocrinology, Leiden University Medical Center, 2300 RC Leiden, The Netherlands; 3Einthoven Laboratory for Experimental Vascular Medicine, Leiden University Medical Center, 2300 RC Leiden, The Netherlands; 4Department of Biomedical Engineering, Eindhoven University of Technology, 5600 MB Eindhoven, The Netherlands; 5Department of Human Genetics, Leiden University Medical Center, 2300 RC Leiden, The Netherlands; 6Laboratory of Experimental Vascular Medicine, University of Amsterdam, Amsterdam UMC, Meibergdreef, 1105 AZ Amsterdam, The Netherlands

**Keywords:** computational modeling, cholesterol, bile acid, energy expenditure, metabolic syndrome, triglycerides, APOE3, CETP

## Abstract

Within the human population, considerable variability exists between individuals in their susceptibility to develop obesity and dyslipidemia. In humans, this is thought to be caused by both genetic and environmental variation. APOE*3-Leiden.CETP mice, as part of an inbred mouse model in which mice develop the metabolic syndrome upon being fed a high-fat high-cholesterol diet, show large inter-individual variation in the parameters of the metabolic syndrome, despite a lack of genetic and environmental variation. In the present study, we set out to resolve what mechanisms could underlie this variation. We used measurements of glucose and lipid metabolism from a six-month longitudinal study on the development of the metabolic syndrome. Mice were classified as mice with either high plasma triglyceride (responders) or low plasma triglyceride (non-responders) at the baseline. Subsequently, we fitted the data to a dynamic computational model of whole-body glucose and lipid metabolism (MINGLeD) by making use of a hybrid modelling method called Adaptations in Parameter Trajectories (ADAPT). ADAPT integrates longitudinal data, and predicts how the parameters of the model must change through time in order to comply with the data and model constraints. To explain the phenotypic variation in plasma triglycerides, the ADAPT analysis suggested a decreased cholesterol absorption, higher energy expenditure and increased fecal fatty acid excretion in non-responders. While decreased cholesterol absorption and higher energy expenditure could not be confirmed, the experimental validation demonstrated that the non-responders were indeed characterized by increased fecal fatty acid excretion. Furthermore, the amount of fatty acids excreted strongly correlated with bile acid excretion, in particular deoxycholate. Since bile acids play an important role in the solubilization of lipids in the intestine, these results suggest that variation in bile acid homeostasis may in part drive the phenotypic variation in the APOE*3-Leiden.CETP mice.

## 1. Introduction

Diets characteristic for Western society have spread across the globe, which together with the development of a mostly sedentary lifestyle, have led to an increase in the prevalence of cardiovascular risk factors, such as obesity, insulin resistance and hypertriglyceridemia [1]. It is generally assumed that the vast differences in the genetic make-up between individuals results in some individuals being less prone and some individuals being more prone to developing these risk factors while being subjected to the same environment. However, genome-wide association studies have to date only been able to explain 21% of the variation in body weight [2]. On the other hand, a significant proportion of the variation in body weight is thought to be due to the variation in environmental variables, such as diet and physical activity. Because of the extreme number of putative variables and the complexity of metabolic control, it has been extremely difficult to sort out the interaction between environmental and genetic factors. Nevertheless, when genes are the dominant drivers of the metabolic syndrome, one would expect inbred animal models housed under standardized conditions to show little variation in phenotype when fed a Western diet. Our recent studies have falsified this hypothesis [3].

Using an apolipoprotein E*3-Leiden (APOE*3Leiden).cholesteryl ester transfer protein (CETP) mouse model ([4]), we showed a human-like variation in phenotypic (obesogenic) response when these mice were fed a Western-type diet. This mouse model is heterozygous for the human APOE*3-Leiden variant, conferring a reduced hepatic uptake of triglyceride-rich lipoprotein remnants from the circulation. Furthermore, the mouse model is heterozygous for human CETP under its endogenous promotor. The combination of these genes results in a ‘humanized’ lipid metabolism with more cholesterol in apoB-containing lipoproteins and a relatively low HDL cholesterol level [4]. 

A striking observation in our previous studies was the major variation in important parameters of the metabolic syndrome, including body weight, plasma triglyceride and cholesterol as well as insulin resistance, not only in time, but also amongst individual mice [3]. Given the fact that the mice are inbred and maintained under identical conditions, this model seems promising to elucidate the mechanism underlying the observed phenotypic variation. Instead of using the usual gene-focused mechanism, we reasoned that, whatever the major source of variation in these animals, the effect must take place through processes involved in energy metabolism. To help us direct our search, we made use of a mixed approach in which computational modeling exploited experimentally obtained longitudinal data to identify the processes most likely involved in explaining the phenotypic differences [5]. Recently, we have published an ordinary differential equation based a computational model called Model INtegrating GLucose and Lipid Dynamics (MINGLeD), which encompasses the main metabolic pathways involved in energy metabolism [6]. In the current study, we concentrated on plasma triglyceride in combination with obesity and used MINGLeD to analyse which processes are altered during the progression of the metabolic syndrome in male APOE*3-Leiden.CETP mice on a high-fat diet. 

## 2. Methods

### 2.1. Animals, Diet and Housing

Experimental conditions have been described previously ([3]). In brief, male APOE*3-Leiden.CETP mice were housed individually and fed a synthetic high-fat and -cholesterol diet (HFCD) containing 60% of energy from fat and 0.25% of weight from cholesterol (D12429, Research Diets) in a light- (lights on 7:00 a.m.–7:00 p.m.) and temperature-controlled (21 °C) facility. At the start of the experimental period, the mice were at the age of 4 months. Prior to the start of the experimental period, mice were co-housed with siblings and fed chow ad libitum. At least one week prior to the start of the experiment, animals were housed individually to acclimatize. Experimental procedures were approved by the Ethics Committees for Animal Experiments of the University of Groningen (Protocol Code 6903).

### 2.2. Experimental Setup

As previously described [3], four groups of mice were fed HFCDs ad libitum for 4 (*n* = 20), 9 (*n* = 19), 13 (*n* = 20) and 28 weeks (*n* = 30), respectively. At the end of the dietary intervention, mice in the respective cohorts were distributed over two groups, to either measure VLDL-TG production or to measure hepatic de novo lipogenesis, measure bile production and collect tissues. In addition, a cohort (*n* = 16) was used to measure endogenous glucose production at weeks 3, 9, 15 and 27 and energy expenditure at weeks 1 and 19. In all groups, blood samples were obtained by tail bleeding at 4- to 6-week intervals, to determine plasma TG, plasma total cholesterol (TC), HDL-C and glucose [3]. In addition, 24 h feces were collected from all groups at 4- to 6-week intervals. Plasma bile acid concentrations were determined by liquid chromatography–tandem mass spectrometry (LC–MS/MS). For quantification, internal standard solution containing D4-cholate, D4-chenodeoxycholate, D4-glycocholate, D4-taurocholate, D4-glycochenodeoxycholate and D4-taurochenodeoxycholate was added to the plasma. Bile salt composition of prepared fecal samples was determined by capillary gas chromatography on an Agilent gas chromatograph (HP 6890). All flux measurements and blood sample collections were started at 1 PM under fasting conditions, with food removed at 9 AM. 

### 2.3. The ADAPT Method

ADAPT is a hybrid modelling method combining data assimilation and machine learning to discover differential equations describing the long-term dynamics of a diet intervention. ADAPT separates the multiscale behavior of metabolic physiology in time. The fast metabolic dynamics (minutes to hours, for example, associated with food intake) are decoupled from the slow dynamics (weeks to months) related to long-term, high-fat and high-cholesterol diet feeding. ADAPT starts from measurements taken at the fast time scale; the slow scale system can be assumed to be in steady state. ADAPT first takes a longitudinal data set and fits a series of polynomial curves and data splines through points sampled from the normal distribution of the data at the respective time points [7]. Subsequently, ADAPT minimizes the error between the model output and the data splines from time point to time point using a least-squares algorithm. Since a penalty is put on changes in the parameter values, the algorithm favors gradual changes in parameters through time over abrupt changes. By studying predicted changes in parameters that were not constrained, ADAPT may assist in identifying processes that are likely to be changed as well, and thus may play an important role in explaining the phenomenon of interest. For the current work, we used 200 time steps and applied a regularization parameter (λ) of 0.01. Further details on how we arrived at the settings used for ADAPT are described in Appendix A (Appendix A).

### 2.4. Experimental Data and Modeling Constraints

The experimental data have been described in detail previously [3]. Some parameters for the assessment of VLDL-TG production, de novo lipogenesis, biliary sterol secretion and liver lipids were obtained cross-sectionally. The number of non-responders in the VLDL-TG production cohort (*n* = 2, *n* = 0, *n* = 0, *n* = 4) and in the cohort undergoing bile cannulation (*n* = 3, *n* = 1, *n* = 1, *n* = 3) for the respective time points of 4, 9, 13 and 28 weeks were too small for reliable differentiation with the responders (*n* = 8, *n* = 9, *n* = 9, *n* = 13 and *n* = 7, *n* = 9, *n* = 8, *n* = 14) in the respective cohorts. Therefore, while constraints for food intake, body weight, plasma parameters and fecal samples were directly taken from the data of responders and non-responders, constraints for liver lipids, biliary secretion, hepatic de novo lipogenesis and VLDL-TG production were taken as group averages and therefore were the same for responder and non-responder groups. Further details as to how experimental data were translated to model constraints may be found in Appendix A. 

### 2.5. Choices Concerning Model Design

We designed a model that includes all fluxes relevant for whole body fat and cholesterol metabolism. Overall, the computational model may be described as a three-link chain in which the metabolic network within a module is more connected than the number of interactions between modules (Figure 1). The model was named Model INtegrating GLucose and Lipid Dynamics (MINGLeD), emphasizing that we have both glucose and lipid metabolism integrated into one model [6]. In general, we considered reactions to be first order. ADAPT uses a data-driven approach to discover a dynamic model. The rate equations reflect the pathway/network structure (connectivity) and stoichiometry and are not based on actual enzyme kinetics. The data-based constraints are used, since ADAPT will change the parameter values to comply with the constraints regardless of the rate equation. All model equations may be found in the Appendix A (Appendix A). We highlight some of the relations here because they require explanation. The rate equation for CETP (j34 in Figure 1) was chosen to be dependent on plasma triglyceride concentration, since this is generally considered to be the driver behind CETP action [8]. The trans-intestinal cholesterol excretion (TICE) rate equation (j37 in Figure 1) was chosen to be dependent on the VLDL-C pool. TICE is the flux of cholesterol that enters the intestine directly from the plasma. The plasma compartments contributing to TICE are not completely clear, and may be both coming from apoB-containing lipoproteins as well as from erythrocytes. Therefore, it was decided to make it dependent on VLDL-C only, since erythrocytes were out of the scope of this study [9].

### 2.6. Validation Experiment

In two institutions (UMCG and LUMC), 13 male APOE*3-L.CETP mice were fed the same HFCD with 60% of energy from fat and 0.25% of weight from cholesterol for 8 weeks and fractional cholesterol absorption was measured as described previously [10]. Animal experiments were approved by the responsible ethics committees. Fecal FFAs were measured as described previously [11].

## 3. Results

### 3.1. Stratification to Responder and Non-Responder Phenotypes

We have shown previously that the APOE*3L.CETP mice show great variability in response to treatments with an HFCD [3]. Data on basic parameters, such as body weight and food intake, are presented in [3]. To be able to differentiate the response of the mice to the HFCD, we stratified the individual mice into the responder and non-responder groups by classifying the animals with a plasma TG < 1.0 mM at the baseline (chow diet) as non-responders. Since non-responders have lower body weights, this also selected mice with lower body weights. This stratification procedure yielded 36 responders and 11 non-responders. As shown in Figure 2, the plasma TG in the non-responding group remained low during the full course of the experiment. In contrast, the plasma TG in the responding mice started to increase at week 4, reached a maximum at week 20 and then decreased sharply to reach a new steady state at week 24. The responding group showed a similar response in the plasma total cholesterol (TC) levels. Interestingly, much less of a difference was observed in the peripheral fat content as well as the plasma HDL-C levels. In agreement with the observed differences in plasma TG and body weight, insulin levels were lower in non-responders as well. (Appendix A).

### 3.2. Application of ADAPT

To address the question which processes are responsible for the observed kinetics in Figure 2, we used the newly developed MINGLeD model of lipid and carbohydrate metabolism [6]. The model contains 18 state variables, 39 parameters and 41 fluxes. In order to find the underlying mechanisms by which the differences between the responders and non-responders can be explained, ADAPT was applied to the MINGLeD model. As depicted in Figure 2, Appendix A, the modeling is able to describe the complex evolution in the measured parameters involved in lipid, glucose and energy metabolism.

### 3.3. ADAPT Predicts Decreased Cholesterol Absorption for Non-Responders

Since non-responders presented with both lower plasma cholesterol values and a higher level of fecal sterol excretion (Figure 2), we inspected which fluxes ADAPT predicted pertaining to cholesterol homeostasis. We then found that ADAPT predicted slightly lower cholesterol absorption for non-responders as compared to responders (Figure 3A), which makes sense in light of the higher observed level of fecal cholesterol excretion and lower plasma TC for the non-responders. While cholesterol synthesis is expected to be decreased with increases in cholesterol absorption [12], no differences in prediction were found for de novo hepatic or peripheral cholesterol syntheses (data not shown).

### 3.4. Validation Experiment of Decreased Cholesterol Absorption

Since ADAPT predicted a lower level of cholesterol absorption in the non-responders compared to responders, we performed a validation experiment in which we measured the cholesterol absorption after feeding the APOE*3-Leiden.CETP mice HFCDs for eight weeks. We reasoned that if the prediction of ADAPT was correct, we would find a positive correlation between the cholesterol absorption and plasma TG. To make sure any effect found would not be site- or cohort-dependent, two independent experiments were performed with different cohorts of mice at two different facilities. As depicted in Figure 3B, this prediction was falsified; no correlation between the cholesterol absorption and plasma TG levels was observed. Furthermore, there was no clear negative correlation between plasma TG and fecal neutral sterol excretion (Appendix A). These findings indicate that cholesterol absorption is not consistently decreased in the non-responder animals.

### 3.5. ADAPT Predicts Higher Glucose Oxidation Rates in Non-responders

Next, we looked for parameter and flux trajectories that may explain the lower body weight observed in the non-responders compared to responders (Figure 2 and Appendix A). Since body weight is the result of the balance between energy absorption and expenditure, any differences must be explained by either. Looking at energy expenditure, we found that ADAPT predicted that non-responders would have higher glucose oxidation rates, while the fat oxidation rates were predicted to be equal between the two groups (Figure 4A,B). This model prediction implies both a higher total energy expenditure for the non-responders, and more glucose stored as fat and enhanced peripheral de novo lipogenesis in the responders. Indeed, if we look at the flux trajectories for these processes, we see that this is also predicted (Appendix A). Since we had put mice in metabolic cages that were monitored at least during the initial period of the experiment, we could compare how the energy expenditure as measured by indirect calorimetry was connected to having a responder or non-responder status. Interestingly, we observed no difference in the energy expenditure between the groups (Figure 4C,D). Of note, even when adjusting for body weight [13], the non-responder animals had a near-average energy expenditure compared to that of the responders.

### 3.6. ADAPT Predicts Lower Fat Absorption in Non-responders

Interestingly, we found that ADAPT predicted a higher level of fat excretion in the non-responders (Figure 5C). This was further highlighted by predictions of a higher TG content in the intestinal lumen and lower parameter values for fat absorption (Figure 5B,D). This prediction was validated by measuring the amount of fatty acids (FFA) still contained in the feces. The FFA content in the feces from the non-responders was indeed higher than that of the responders, suggesting impaired fat absorption in these mice (Figure 6A). While the cumulative fecal fat excretion was significantly different between the groups, not all the non-responders presented with an increased level of fecal FFA excretion, suggesting that, in these animals, the plasma TG is low for another reason. Interestingly, the fecal FFA excretion also negatively correlated with the body weight and plasma TG (Figure 6E,F). However, the correlation between the body weight and fecal FFA excretion was much more evident than that for the plasma TG, whose additional variation obviously must be from another factor.

### 3.7. Decreased Fat Absorption Is Associated with a Lower Hydrophobicity Index of Fecal Bile Acids

Since cholesterol absorption and fat absorption are more promoted by hydrophobic than hydrophilic bile acids [13,14,15], we compared the bile acid composition profiles in the feces, plasma and bile between the responders and non-responders. We reasoned that the higher observed level of fecal FFA excretion may be related to the hydrophobicity of bile acids. Indeed, regardless of responder or non-responder statuses, the hydrophobicity index of the fecal bile acids was correlated with the fecal FFA excretion (Figure 6C). Furthermore, the fecal hydrophobicity index was positively associated with the fecal bile acid excretion as well. In fact, we found fecal FFA excretion to be more strongly correlated with fecal bile acid excretion than with the hydrophobicity index (Figure 6D). Interestingly, we found that the fecal deoxycholic acid was especially highly correlated with the fecal FFA (Appendix A). Surprisingly, comparing the biliary bile acid profiles of the responders from all time points with those of the non-responders, neither a difference in the total biliary bile acid secretion nor in the individual bile acids was found (Appendix A). However, there was a trend of a lower hydrophobicity index of the biliary bile acids for the non-responders (*p* = 0.12). Interestingly, when only the biliary bile acid profiles of the first three months were compared, when the level of fat excretion was the highest, the hydrophobicity index of the biliary bile acids was indeed lower in the non-responders (*p* = 0.001). Moreover, the mice with low plasma TG, in which the fractional cholesterol absorption was measured after eight weeks of the HFCDs, also showed biliary bile acid profiles with lower hydrophobic indexes than those with higher plasma TG (*p* = 0.01). Furthermore, both cholic acid- and chenodeoxycholic acid-derived bile acids were higher in the plasma of the non-responders (Appendix A). Together, these data suggest that the difference in the fecal FFA excretion in non-responders is driven by changes in the bile acid metabolism.

## 4. Discussion

The major result of this study is that by using the computational modeling method ADAPT, it is possible to analyze in detail the pathways that induce the progression of comorbidities in complex diseases. In this study, we applied the method to explain the phenotypic variation induced by long-term HFCDs in APOE*3-Leiden.CETP mice. The application of ADAPT for adjusting parameters in the MINGleD model allowed an accurate modelling of the phenotypic changes in long-term experiments. The ADAPT analysis suggested different pathways, such as decreased cholesterol absorption, increased energy expenditure and an increased level of fecal fat excretion to explain the phenotypic variability in body, hepatic and plasma TG. Subsequent validation experiments failed to confirm a decreased cholesterol absorption and an increased energy expenditure to explain the lack of response in a subset of the mice. In contrast, an increased level of fecal fat excretion in the non-responders could be confirmed. Furthermore, we found that the increased level of fecal fat excretion was associated with a decreased level of fecal bile acid excretion, suggesting that a decrease in bile acid production may, at least in part, drive the lower body weight and plasma TG in non-responders. A similar relation between deoxycholic acid and body weight loss was reported by [16] However, they observed the interaction of intestinal fat with GPR119, causing an increased satiety signal. In our study, we did not observe a decreased level of food intake in the non-responder mice.

### 4.1. Hybrid Modeling Using ADAPT

ADAPT is a hybrid modelling method that employs the power of data-driven and mechanistic modelling techniques to estimate the long-term dynamics of metabolic physiology. ADAPT uses concepts somewhat similar to physics-informed neural networks (PINNs) and other data assimilation methods emerging in cardiovascular modeling [17]. The network structure of the metabolic system imposes strong constraints on the solution space of the mathematical model, which ADAPT combines into equations for kinetics and fluxes with time-series data. The trade-off between the bias variance is controlled by the hyperparameter λ. Lambda was tuned to provide enough flexibility to discover possible explanations underlying the phenotype heterogeneity in APOE*3-Leiden.CETP mice induced by long-term HFCDs (Appendix A). For higher values of λ, the fluctuations in the model dynamics could be dampened, but with too high values of λ, the goodness of fit decreases and a bias emerges. The stochastic data model and spline interpolation to take into account experimental and biological uncertainties also contribute to the variation in model predictions. Other types of data, e.g., transcriptomics, can also be included in ADAPT to further constrain the model predictions. Gene expression data have been incorporated by expanding the regularization function in ADAPT [18].

### 4.2. Fat Absorption and Energy Expenditure

In line with the major impact of fat absorption in the model, we found that the non-responders are marked by an increased level of fecal fat excretion. However, while the cumulative fat excretion varied between 0.5 and 5 g, the body weight difference amounted up to 36 g, and is thus roughly ten times as large. This indicates that any differences in absorption must be accompanied by a difference in energy expenditure. In fact, we predicted that the energy expenditure in the non-responders is increased compared to the responders, which ADAPT mainly attributed to a difference in glucose oxidation. Indirect calorimetry, however, showed no statistical significant differences in the respiratory ratio (RER) or increases in energy expenditure. Our results are in agreement with findings of Tarasco et al., who failed to find differences in energy expenditure between responder and non-responder APOE*3-Leiden.CETP mice as well [19].

A possible explanation is that indirect calorimetry may not be sensitive enough to detect the difference in energy expenditure between the responders and non-responders. The mean difference in the weight increase between non-responders and responders in the first 12 weeks was 5 g. This difference in weight would amount to an energy imbalance, presuming the weight difference is on account of fat (0.6 kcal/day), which, assuming a daily energy expenditure of 12 kcal/day, would be 5% of the energy expenditure. Coincidentally, 5% of the energy expenditure is about the threshold to detect differences in energy expenditure using indirect calorimetry [20,21]. Furthermore, while a two-fold higher glucose oxidation rate was predicted for the non-responders compared to the responders, the RER data from the non-responders are not in agreement with this prediction. However, it should also be considered that, given the number of responders and non-responders and the expected difference in RER if the glucose utilization was 10% for the responders and 20% for the non-responders, the power to detect this difference would be less than 50%. All in all, despite a lack of observed differences using indirect calorimetry, the non-responders likely have an increased level of energy expenditure in addition to a decreased level of fat absorption.

It is tempting to speculate that a decrease in fat absorption also leads to a higher level of energy expenditure. While the mechanism behind this is unclear, it has been proposed that this effect may be due to a shift towards absorption more distally in the small intestine, leading to less chylomicron production and with a smaller size. These smaller chylomicrons are then believed to tip the scale more towards utilization than storage, explaining the higher energy expenditure [22]. In humans, bariatric surgery also leads to increased nutrient availability in the distal small intestine. The weight loss associated with bariatric surgery, however, is neither due to the increased energy expenditure nor malabsorption per se, but attributable to the decrease in food intake in response to the increased production of incretins, such as GLP-1 [23,24]. Since in this study no decrease in food intake was observed (Appendix A), such a mechanism is likely not relevant here.

### 4.3. Bile Acid Metabolism

An important result of this study was that apart from the tight association in male APOE*3-Leiden.CETP mice between body weight and fecal fat excretion, there is also a tight connection between fecal fat excretion and bile acid metabolism. However, what drives the observed differences in bile acid homeostasis remains unclear. Tarasco et al. reported that the livers of non-responder APOE*3-Leiden.CETP mice were found to have more inflammation and less steatosis, and occasionally formed neoplasms [19]. In the current study, though no significantly increased inflammation was found, we did observe less steatosis in non-responder mice, and found (pre)neoplastic deformations in two out of the nine non-responder mice whose histology was available. However, the difference in the response to the HFCDs was present from the moment the TG levels rose in the responding mice, probably in part caused by the difference in bile acid excretion which was much lower in the non-responding mice (Figure 2) at the onset of the experiments. Together this makes liver injury as an explanation for the difference in bile acid homeostasis not very likely. We conclude that the altered fecal bile acid profile and concurrent changes in the fat uptake dynamics may contribute substantially to the observed decrease in the plasma lipids and the lower body weight found in the non-responders. While the correlation between the fecal deoxycholic acid level and body weight was strong, it should be noted that deoxycholic acid is mainly produced in the colon and is therefore not likely to significantly contribute to fat uptake in the small intestine. Rather, fecal deoxycholic acid is likely a sensitive marker for the availability of cholic acid in the small intestine, through the interaction with the microbiome [25,26]. Cholic acid in the small intestine then drives the observed fat uptake dynamics. The question arises whether our results in a mouse with a humanized lipid profile can be translated to development of the metabolic syndrome in humans. The efficiency of fat absorption in humans is comparable to that in mice, hence a small decrease may also substantially affect the progression of obesity in humans. Although bile acid homeostasis differs substantially between mice and humans, in both species the level of the excretion of deoxycholic acid is high, hence similar mechanisms could play a role. The pharmacological perturbation of bile acid metabolism may prove successful in combating obesity and dyslipidemia in humans.

## 5. Conclusions

This study demonstrates how a systems analysis may be used to explore heterogeneity in the propensity to develop the metabolic syndrome. Using ADAPT, we show that there is increased level of fecal fat excretion and that there must be an increased level of energy expenditure in APOE*3-Leiden.CETP-mice that do not respond to a HFCD. Finally, we show that these differences appear to be coupled to a decreased production of bile acids and a decrease in the fecal excretion of deoxycholic acid. Further studies should address whether similar mechanisms may be responsible for the differences in susceptibility in developing dyslipidemia and obesity in the human population.

## Figures and Tables

**Figure 1 nutrients-14-04936-f001:**
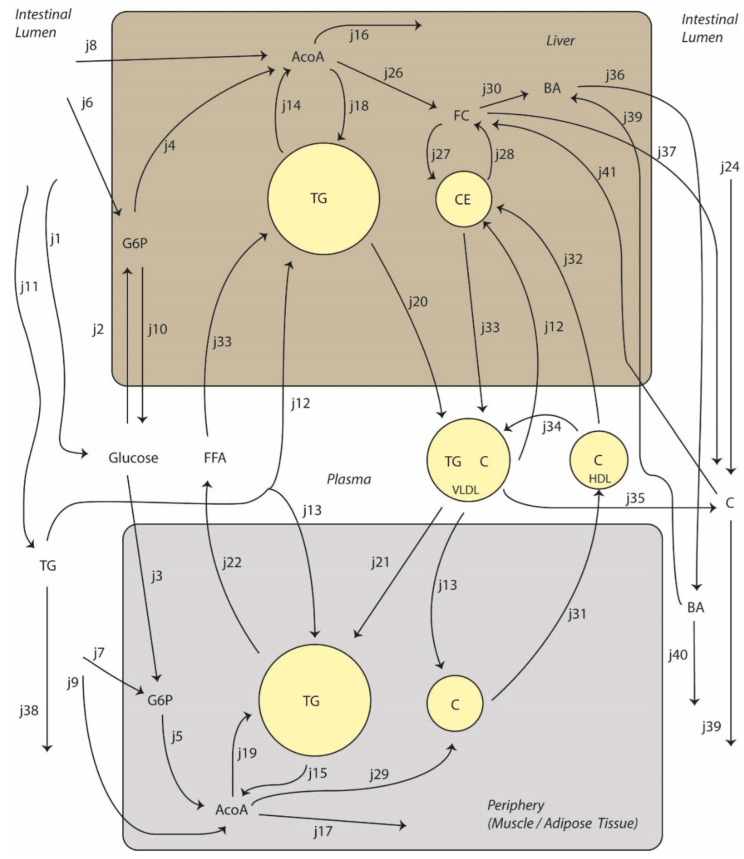
Schematic of the MINGLeD model. The MINGLeD model consists of four compartments (liver, plasma, periphery and intestinal lumen), 18 states and 41 fluxes. Food intake is modeled as glucose entering the plasma (j1), triglyceride (TG, j11) and cholesterol (C, j24) entering the intestinal lumen, whereas amino acids from protein are distributed to liver and periphery at the level of glucose-6-phosphate (G6P, gluconeogenic) or acetyl-CoA (AcoA, ketogenic) (j6, j7, j8, j9). Glucose in the plasma is absorbed by liver (j2) and periphery (j3) to enter the Krebs cycle (j16, j17) or to be used for biosynthetic processes, such as de novo lipogenesis (j19) or cholesterol (j26, j29) and bile acid synthesis (j30). TG from the intestinal lumen can be absorbed by the liver (j12) or periphery (j13) and be used for beta-oxidation (j14, j15) or redistribution as VLDL (j20) or free fatty acids (FFA, j22). Absorbed dietary cholesterol first enters the liver (j41) where it can be used for bile acid synthesis (j30) or redistributed to the periphery in the form of VLDL-C (j33, j13). Peripheral cholesterol pools can return to the liver through HDL-C (j31, j32) or VLDL-C after action of cholesteryl ester transfer protein (j34, j12). Cholesterol can be cleared from the body through biliary cholesterol secretion (j37) or trans-intestinal cholesterol secretion (j35).

**Figure 2 nutrients-14-04936-f002:**
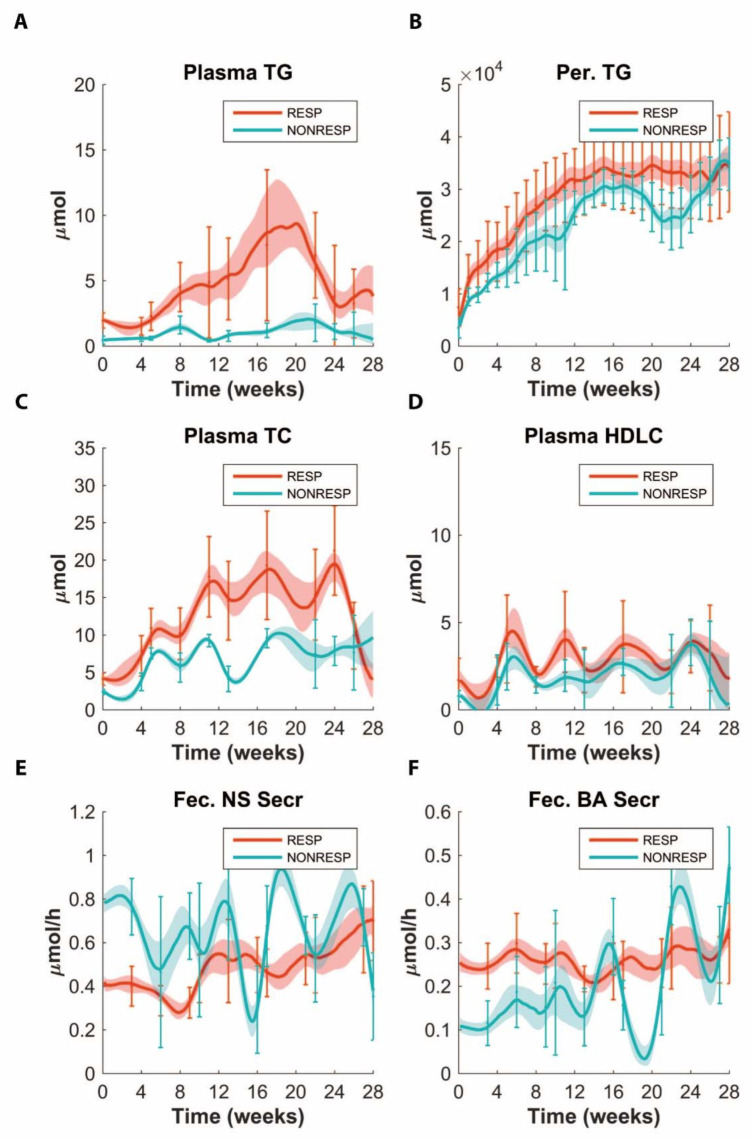
Plasma TG (**A**) and peripheral fat (Per. TG (**B**)), plasma total cholesterol (TC) (**C**), HDL cholesterol (HDLC) (**D**), fecal neutral sterol secretion (**E**) (Fec. NS Secr) and fecal bile acid secretion (**F**) (Fec. BA Secr) for responders (RESP) and non-responders (NONRESP), respectively, with their respective fits in the ADAPT model simulation. Note that non-responders are marked by lower plasma TG and less peripheral fat. Error bars represent data with standard deviation, bold lines represent the median solution of all ADAPT simulations and the areas represent 30% around the median solution.

**Figure 3 nutrients-14-04936-f003:**
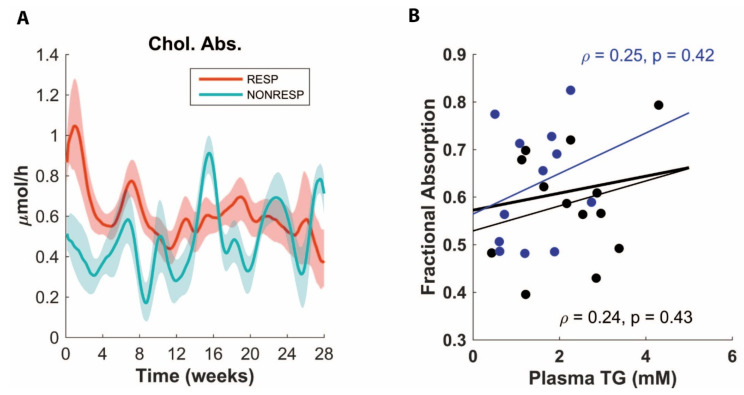
Cholesterol absorption (Chol. Abs.) as predicted by ADAPT (**A**) for responders (RESP) and non-responders (NONRESP); note that ADAPT predicts lower level of cholesterol absorption for non-responders. Fractional cholesterol absorption (**B**) in two cohorts (black and blue) of mice after 8 weeks of HFCD. Note there is no correlation between fractional cholesterol absorption and plasma TG.

**Figure 4 nutrients-14-04936-f004:**
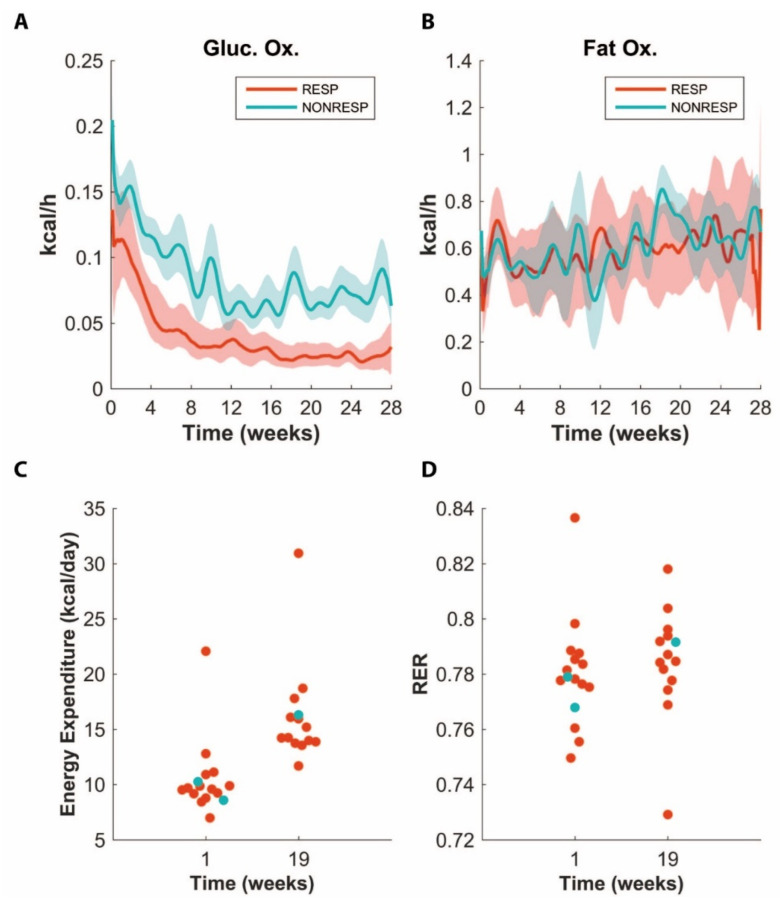
Predictions for glucose oxidation (Gluc. Ox.) (**A**) and fat oxidation (Fat Ox.) rate (**B**) in responders and non-responders, respectively. The line represents the median values, whereas the area around the line denotes 30% of solutions around the median. Energy expenditure (**C**) and respiratory exchange ratio (RER) (**D**) for animals after 1 and 19 weeks of HFCDs. Note how the non-responders (blue) are not necessarily marked by increased energy expenditure.

**Figure 5 nutrients-14-04936-f005:**
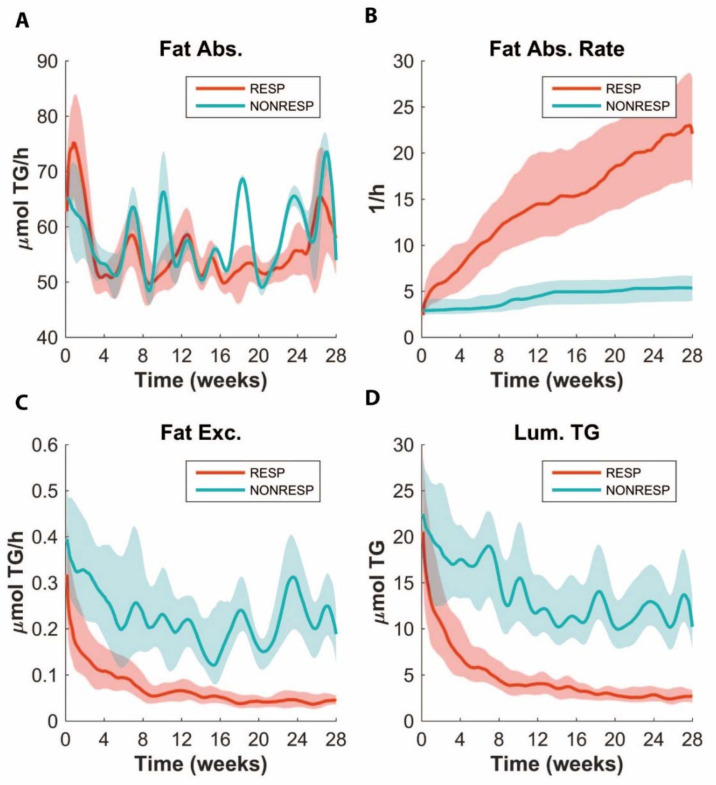
Predictions for fat absorption (Fat Abs.) (**A**), fat absorption rate (Fat Abs. Rate) (**B**), fecal fat excretion (Fat Exc.) (**C**) and intestinal lumen fat content (Lum. TG) (**D**) in responders (RESP) and non-responders (NONRESP), respectively. The bold line represents the median values, whereas the area around the line denotes 30% of solutions around the median. Note how the fat absorption rate is predicted to be slower, while fat excretion and intestinal fat content are predicted to be increased in non-responders.

**Figure 6 nutrients-14-04936-f006:**
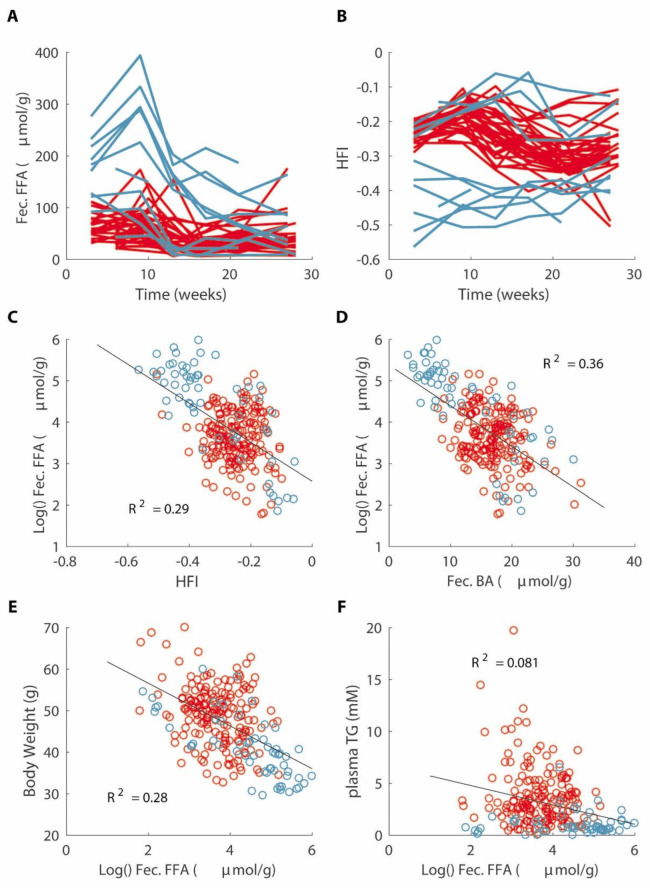
Fecal fatty acid excretion (Fec. FFA) (**A**) and hydrophobicity index (HFI) of fecal bile acids (**B**) over time. Correlations between fecal fatty acid excretion and HFI (**C**), fecal bile acids (Fec. BA) (**D**), body weight (**E**) and plasma TG (**F**). Responders are marked in red and non-responders are marked in blue.

## Data Availability

Data have been published in part in [3] and Appendix A.

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
