# Peer review of "A Systems Analysis of Phenotype Heterogeneity in APOE*3Leiden.CETP Mice Induced by Long-Term High-Fat High-Cholesterol Diet Feeding"

_nutrients, 2022, doi:10.3390/nu14224936_

Round 1

Reviewer 1 Report

The manuscript entitled "A systems analysis of phenotype heterogeneity in AP-2 OE*3Leiden.CETP mice induced by long term high-fat high-3 cholesterol diet feeding" is an interesting work in which the authors apply systematic analysis tools to identify biological markers to explain the variability between subjects in the development of obesity as a risk factor for other diseases.  It is a topical, novel approach that could provide information to understand the mechanisms underlying the development of metabolic syndrome and obesity. However, there are some aspects that the authors should assess.

Introduction

- The authors should be specific about the aim of the paper, and the last lines of the introduction should be devoted to this purpose. After reading the introduction, it is confusing to understand exactly what the authors intend to do with the paper.

- This reviewer understands that lines 88-93 are a summary of the results of the paper, it is suggested that it be removed and used to support the reader in the first few lines of the discussion.

Methods

- Line 111. The authors should clarify why the experiment on glucose production and energy expenditure is carried out on 16 individuals and not on 20 as in the other 4 groups. Why is this group independent of the other 4 groups?

- Line 240. The authors should clarify why the validation trial is conducted with 8 weeks follow-up and not 28 weeks? The outcome may be affected by the follow-up period?

-Results

- Line 197. Could the authors explain why 0.1 mM is established as the cut-off point for categorising individuals according to triglycerides? Are there any previous studies that support this?

- Figura 2A. Can the authors explain why trglycerides in responder mice decrease after week 20 and should not continue to increase?

- In line 204 the authors indicate that total cholesterol (TC) and glucose follow a similar profile.  However, in figure 2 the TC profile is shown but the glucose profile is not.  Glucose and TC are shown in supplementary figure S1 but do not follow the same profile. This is confusing please clarify.

- Figure 3A. Are there statistically significant differences in cholesterol absorption profiles between responders vs. non-responders?  Although the ADAP analysis suggests that cholesterol absorption is slightly lower in non-responders compared to responders, this is apparently not statistically significant. This suggests that the validation experiment is not necessary, where indeed cholesterol absorption is not consistently lower in non-responders, and should be eliminated.

- The authors could provide a figure or table showing the mean weight of the responder vs. non-responder group at each follow-up point.

- The authors should apply statistical tests to validate that there are no significant differences in energy expenditure between responders vs. non-responders in the intervention period.

Discussion

- The authors should consider in the discussion section how to apply the observed knowledge to human health, i.e. to enhance the applicability of the results to control or reduce the risk of developing metabolic syndrome and/or obesity.  Are there previous studies in humans suggesting a relationship between intestinal fatty acids and bile acid levels in subjects with metabolic syndrome?

Author Response

In this study, the authors ecplored which processes are altered during progression of metabolic syndrome in male APOE*3-Leiden.CETP mice on a high fat diet. They found that there is increased fecal fat excretion and that there must be increased energy expenditure in APOE*3-Leiden.CETP-mice that do not respond to a HFCD. The variation in lipid metabolism and bile acid homeostasis may drive the phenotypic variation.

Specific comments

The animal models used, what specific characteristics are similar to humans obesogenic response. If the ADAPT method Can be used to predict obesity and dyslipidemia progression in humans.

While mice carry most of their plasma cholesterol in high-density lipoprotein (HDL), humans have a large LDL-c fraction because of the presence of cholesteryl ester transfer protein (CETP). To mitigate these species-specific differences, we made use of ‘humanized’ apoE3*L.CETP mice, in which the apoE3*L transgene confers reduced clearance of chylomicron-remnants, and the CETP transgene results in a cholesterol profile more closely resembling that of humans. On a high fat diet these mice readily develop obesity (Westerterp et al. ATVB 2006; 26: 2552-2559). Interestingly, in contrast to other mouse models of atherosclerosis, the apoE3*L.CETP mouse has been shown to respond similar to treatment with statins, fibrates and ezetimibe as humans in terms of changes in plasma cholesterol and atherosclerotic progression (Van de Hoek et al DOM 2014 16: 537-544).

Methods for the determination and calculation of bile acids should be described.

We have adapted the methods section to describe more clearly methods for quantitative bile acid measurement in plasma and feces.

Fig 1. The lines are a little messy. It is recommended to stratify according to the degree of correlation.

We agree that the number of lines in Fig.1 is substantial. However, we feel it is important to show that we have incorporated a great number of reactions in the model. In our view this enables readers to grasp the specifics of our modeling approach.

Basic physical characteristics of the animal, such as body weight and food intake, should be described.

The basic physical parameters have been published in the phenomenological report of this study. We now refer more directly to this paper in line 2-3 of the Results section.

Dissussion Please comment on the differences of fat absorption and energy expenditure between mice and humans.

We have added the following sentences to the discussion section to clarify this point.

“The question arises whether our results in a mouse with a humanized lipid profile, can be translated to development of metabolic syndrome in humans. The efficiency of fat absorption in humans is comparable to that is mice, hence a small decrease may also substantial affect progression of obesity in humans. Although bile acid homeostasis differs substantially between mice and humans in both species excretion of deoxycholic acid is high, hence similar mechanisms could play a role. Pharmacological perturbation of bile acid metabolism may prove successful in combating obesity and dyslipidemia in humans.”

Reviewer 2 Report

In this study, the authors ecplored which processes are altered during progression of metabolic syndrome in male APOE*3-Leiden.CETP mice on a high fat diet. They found that there is increased fecal fat excretion and that there must be increased energy expenditure in APOE*3-Leiden.CETP-mice that do not respond to a HFCD. The variation in lipid metabolism and bile acid homeostasis may drive the phenotypic variation.

Specific comments

The animal models used, what specific characteristics are similar to humans obesogenic response. If the ADAPT method Can be used to predict obesity and dyslipidemia progression in humans.

Methods for the determination and calculation of bile acids should be described.

Fig 1. The lines are a little messy. It is recommended to stratify according to the degree of correlation.

Basic physical characteristics of the animal, such as body weight and food intake, should be described.

Dissussion Please comment on the differences of fat absorption and energy expenditure between mice and humans.

Author Response

The manuscript entitled "A systems analysis of phenotype heterogeneity in AP-2 OE*3Leiden.CETP mice induced by long term high-fat high-3 cholesterol diet feeding" is an interesting work in which the authors apply systematic analysis tools to identify biological markers to explain the variability between subjects in the development of obesity as a risk factor for other diseases.  It is a topical, novel approach that could provide information to understand the mechanisms underlying the development of metabolic syndrome and obesity. However, there are some aspects that the authors should assess.

Introduction

- The authors should be specific about the aim of the paper, and the last lines of the introduction should be devoted to this purpose. After reading the introduction, it is confusing to understand exactly what the authors intend to do with the paper.

We have changed the last part of the introduction to put more emphasis on the main aim of the study

- This reviewer understands that lines 88-93 are a summary of the results of the paper, it is suggested that it be removed and used to support the reader in the first few lines of the discussion.

We have deleted this section of the introduction and adapted the last sentences to clarify the aim of this study

Methods

- Line 111. The authors should clarify why the experiment on glucose production and energy expenditure is carried out on 16 individuals and not on 20 as in the other 4 groups. Why is this group independent of the other 4 groups?

Because of the limited availability of metabolic cages to measure energy expenditure this experiment was carried out in a smaller group of mice.

- Line 240. The authors should clarify why the validation trial is conducted with 8 weeks follow-up and not 28 weeks? The outcome may be affected by the follow-up period?

-Results

- Line 197. Could the authors explain why 0.1 mM is established as the cut-off point for categorising individuals according to triglycerides? Are there any previous studies that support this?

The mice were bred by the group of Professor Rensen at the Leiden University Medical Center, Leiden. The cut-off point of <1 mM triglycerides at baseline has been established by his group

- Figura 2A. Can the authors explain why trglycerides in responder mice decrease after week 20 and should not continue to increase?

Responder mice indeed show a decrease in TG after week 20. We have carried out multiple experiments to investigate the mechanism but did not succeed in explaining this phenomenon.

- In line 204 the authors indicate that total cholesterol (TC) and glucose follow a similar profile.  However, in figure 2 the TC profile is shown but the glucose profile is not.  Glucose and TC are shown in supplementary figure S1 but do not follow the same profile. This is confusing please clarify.

The reviewer is correct. Glucose is not shown in figure 2 and does not show the same time course of the biphasic response. We meant here to say that also glucose shows a biphasic response. We apologize for the confusion generated and have changed the text.

- Figure 3A. Are there statistically significant differences in cholesterol absorption profiles between responders vs. non-responders?  Although the ADAP analysis suggests that cholesterol absorption is slightly lower in non-responders compared to responders, this is apparently not statistically significant. This suggests that the validation experiment is not necessary, where indeed cholesterol absorption is not consistently lower in non-responders, and should be eliminated.

The reviewer is correct. Using mixed linear modeling statistics we could not find statistical significant differences in cholesterol absorption. Since ADAPT analysis did predict a difference we validated this result in a separate experiment and had to conclude that the prediction was incorrect.

- The authors could provide a figure or table showing the mean weight of the responder vs. non-responder group at each follow-up point.

We felt to generate more insight between responders and non responders by showing the individual data of the mice. These are depicted in supplemental figure 1, red symbols for responders; blue for non-responders.

- The authors should apply statistical tests to validate that there are no significant differences in energy expenditure between responders vs. non-responders in the intervention period.

Apologies for the omission but we did test significance of changed energy expenditure but could not find this. We now added this to the discussion. Note that also Tarrasco et al (2018) did not find differences in energy expenditure.

Discussion

- The authors should consider in the discussion section how to apply the observed knowledge to human health, i.e. to enhance the applicability of the results to control or reduce the risk of developing metabolic syndrome and/or obesity.  Are there previous studies in humans suggesting a relationship between intestinal fatty acids and bile acid levels in subjects with metabolic syndrome?

To our knowledge the relation between bile acid and fatty acid excretion on the one hand and control of body weight on the other hand has never been studied in humans.  To discuss better the option of translating our results from mice to humans we have added the following sentences in the discussion section. “The question arises whether our results in a mouse with a humanized lipid profile, can be translated to development of metabolic syndrome in humans. The efficiency of fat absorption in humans is comparable to that is mice, hence a small decrease may also substantial affect progression of obesity in humans. Although bile acid homeostasis differs substantially between mice and humans in both species excretion of deoxycholic acid is high, hence similar mechanisms could play a role. Pharmacological perturbation of bile acid metabolism may prove successful in combating obesity and dyslipidemia in humans.”